# Trial of Chemical Composition Estimation Related to Submarine Volcano Activity Using Discolored Seawater Color Data Obtained from GCOM-C SGLI. A Case Study of Nishinoshima Island, Japan, in 2020

Yuji Sakuno

Graduate School of Advanced Science and Engineering, Hiroshima University,
Higashi-Hiroshima 739-8527, Japan; sakuno@hiroshima-u.ac.jp; Tel.: +81-82-424-7773

**Abstract:** This study aims to develop the relational equation between the color and chemical composition of discolored seawater around a submarine volcano, and to examine its relation to the volcanic activity at Nishinoshima Island, Japan, in 2020, using the model applied by atmospheric corrected reflectance 8 day composite of GCOM-C SGLI. To achieve these objectives, the relational equation between the RGB value of the discolored seawater in the submarine volcano and the chemical composition summarized in past studies was derived using the XYZ colorimetric system. Additionally, the relationship between the volcanic activity of the island in 2020 and the chemical composition was compared in chronological order using the GCOM-C SGLI data. The following findings were obtained. First, a significant correlation was observed between the seawater color (x) calculated by the XYZ colorimetric system and the chemical composition such as (Fe + Al)/Si. Second, the distribution of (Fe + Al)/Si in the island, estimated from GCOM-C SGLI data, fluctuated significantly just before the volcanic activity became active (approximately one month prior). These results suggest that the chemical composition estimation of discolored seawater using SGLI data may be a powerful tool in predicting submarine volcanic activity.

**Keywords:** submarine volcano; GCOM-C SGLI; discolored seawater





## 1. Introduction

In recent years, there have been frequent eruptions of submarine volcanoes. Anak Krakatau in Indonesia [1], White Island in New Zealand [2], and Nishinoshima Island in Japan [3] are just three of the submarine volcanoes that have erupted in the past 2 years; Nishinoshima Island, particularly, has had active volcanic activity since December 2019. Such eruptions of submarine volcanoes not only hinder the navigation of ships and aircraft in the area, but can also have other life-threatening implications. Hence, it is extremely important to monitor these volcanoes to be able to predict any volcanic activity. Scarpa and Tilling [4] have evaluated various methods related to volcano observation, noting the importance of monitoring volcanic earthquakes, the expansion of magma pools, increases in volcanic gas release, and rises in temperature. However, considering that it is dangerous to conduct such surveys, remote sensing surveys have often been used instead.

Remote sensing volcanic surveys include topographic analysis of both volcanic islands, using satellite Synthetic Aperture Radar (SAR) data [5,6], and mountain bodies, using thermal infrared sensors (thermography) mounted on helicopters, aircraft, and satellites, as well as temperature analysis [7,8], silicon dioxide ($SiO_2$) analysis of volcanic gas using satellite infrared sensors [9,10], and eruption analysis using meteorological satellite visible and infrared sensors [11,12]. Although the relationship between the chemical composition of discolored seawater and volcanic activity has been known for a long time [13,14], there have been very few quantitative studies of the chemical composition itself that have been

conducted using remote sensing, and in the few studies that have been undertaken [15,16], only the reflectance pattern of discolored seawater has been analyzed.

Submarine volcanoes release several chemicals, including iron (Fe), aluminum (Al), and silicon (Si), depending on their activity, and these chemicals change the color of the surrounding sea. Generally, a higher proportion of Fe produces a yellow or brown color, whereas a higher proportion of Al or Si produces a white color [17]. Focusing on this discoloration phenomenon, Watanabe [18] recently attempted to use previously taken aerial photographs to quantify Fe, Al, and Si from the RGB values of these images. However, there was found to be no significant correlation between the RGB values and the absolute values of the concentration of these chemical substances given that "the RGB values of the analyzed aerial photographs were not removed from the influence of brightness due to the shooting time or the weather" and that "volcanic discolored seawater is determined not by the absolute values of Fe, Al, and Si, but by the relative mixing ratio".

However, there have been a greater number of studies that have used satellites to chromatically analyze hot spring water, instead of discolored seawater. Ohsawa et al. [19] and Onda et al. [20] have explained the coloration mechanism of water in typical hot springs in Japan using the chromaticity diagram of the XYZ colorimetric system [21]. They identified that the blue to green colors of hot spring water and crater lakes are scattered not only by the sulfur colloid, but also by the absorption of dissolved $Fe^{2+}$ sunlight.

In recent years, satellites such as Terra/Aqua Moderate Resolution Imaging Spectroradiometer, Sentinel-3 Ocean and Land Color Imager (OLCI), and Global Change Observation Mission–Climate Second Generation Global Imager (GCOM-C SGLI) have been the typical global satellites used in measuring the color of seawater. Among them, the SGLI sensor, launched by Japan in December 2017, is the seawater color sensor that has the highest resolution (250 m), and it has an observation cycle of 2–3 days. In conventional satellite analysis of discolored seawater, satellites with a period of 16 days or more, such as Terra Advanced Spaceborne Thermal Emission and Reflection Radiometer (ASTER) and Advanced Land Observing Satellite Advanced Visible and Near-Infrared Radiometer type 2 (ALOS AVNIR-2), have typically been used [15,16]. However, given the life-saving potential of being able to predict volcanic eruptions, monitoring using satellites with the shortest possible cycle is much preferred.

It is within this context that this study has been undertaken with the objectives of developing the relational model between the color of seawater and the chemical composition around submarine volcano using the XYZ colorimetric system, and examining the volcanic activity of Nishinoshima Island in 2020 using GCOM-C SGLI data.

## 2. Materials and Methods

### 2.1. Study Area

Nishinoshima Island is located in the Ogasawara Arc, at 27°15′ N and 140°52.5′ E, approximately 1000 km south of Tokyo, Japan, as shown in Figure 1. The eruption started in November 2013, after 40 years of dormancy, and continued until November 2015, generating lava of approximately $8.7 \times 10^7$ m$^3$ above sea level, which increased the size of the island, making it approximately 13 times larger [22]. The most recent eruption began in December 2019, with further eruptions continuing until mid-August, after which the eruptive activity subsided [23]. As of December 2020, the island is approximately 2 km$^2$ in size. Throughout 2020, yellow-brown, yellow-green, and brown discolored seawaters were regularly observed off the coast of Nishinoshima, as can be seen in the recent Landsat-8 image shown in Figure 1.

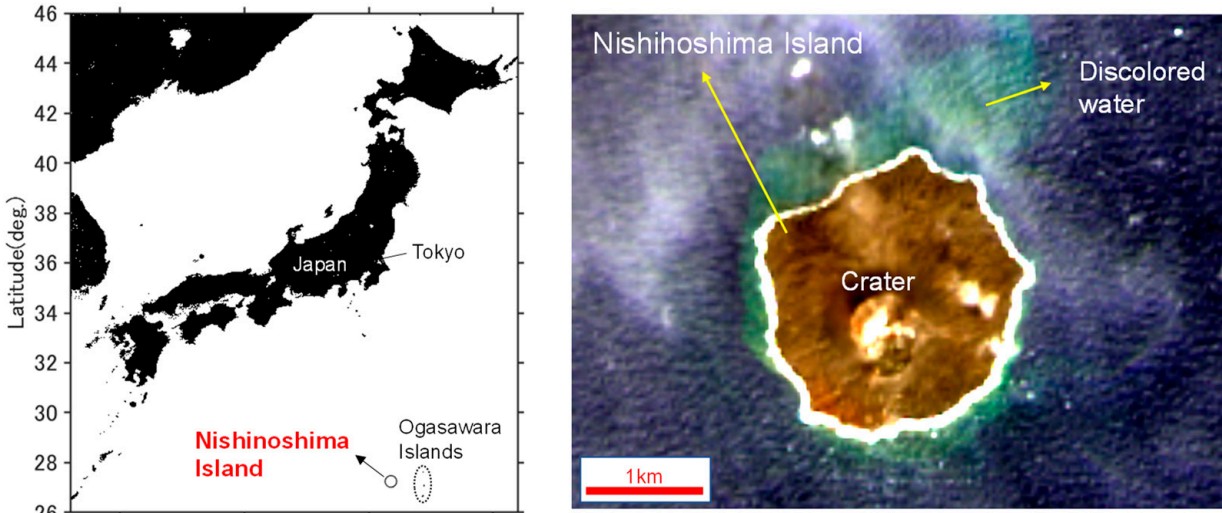

**Figure 1.** Map of Nishinoshima Island (**left**) and Landsat-8 Operational Land Imager (OLI) image of the island taken on 5 September 2020 (**right**).

### 2.2. XYZ Colorimetric System Conversion from RGB and Reflectance Data

The color system used in this study is the XYZ colorimetric system by the Commission internationale de l'éclairage (CIE) in 1931 [21,24,25], which has a proven track record in the color evaluation of crater lakes and hot spring water [19,20]. This XYZ colorimetric system has the advantage of being able to express the reflectance obtained from the RGB information of aerial photographs and satellite data in the same color space. First, the following equation [24] was used as a formula for converting the RGB values obtained from aerial photographs taken from previous studies into the XYZ colorimetric system:

$$\begin{pmatrix} X \\ Y \\ Z \end{pmatrix} = \begin{pmatrix} 0.4125 & 0.3576 & 0.1804 \\ 0.2127 & 0.7152 & 0.0722 \\ 0.0193 & 0.1192 & 0.9503 \end{pmatrix} \begin{pmatrix} R \\ G \\ B \end{pmatrix} \tag{1}$$

where, R, G, and B in Equation (1) are values that have been converted into 0–1 relative values by dividing the 0–255 (8-bit spectral) values obtained from the aerial photograph image by 255.

The following equation [19] was used to convert the reflectance into the XYZ values (X, Y, and Z) of the XYZ colorimetric system.

$$X = K \int_{380}^{780} S(\lambda) x\prime(\lambda) R(\lambda) d\lambda \tag{2}$$

$$X = K \int_{380}^{780} S(\lambda) y\prime(\lambda) R(\lambda) d\lambda \tag{3}$$

$$X = K \int_{380}^{780} S(\lambda) z\prime(\lambda) R(\lambda) d\lambda \tag{4}$$

where $S$ is the spectral distribution of standard light; $x'$, $y'$, and $z'$ are the color matching function; $R$ is the spectral reflectance of the object; $K$ is the proportional coefficient (although, it is not necessary to consider this because of the relativizing used in this study); and $\lambda$ is the wavelength. The chromaticity coordinates x, y, and z are derived by normalizing the values of X, Y, and Z as the following equations:

$$x = \frac{X}{X + Y + Z} \tag{5}$$

$$y = \frac{Y}{X + Y + Z} \tag{6}$$

$$z = \frac{Z}{X + Y + Z} \tag{7}$$

Since the variables x, y, and z in these equations have a relationship of x + y + z = 1, z is automatically determined if $x$ and $y$ are known. Hence, in principle, if an $R(\lambda)$ of 380–780 nm is measured, which is commonly included in Equations (2)–(4), the remaining coefficients can be calculated using constants such as the JIS Handbook [25]. The discolored seawater can be quantified (standardized) by the two-dimensional chromaticity coordinates of $x$ and $y$. However, for $S(\lambda)$, various light sources have been proposed depending on the purpose. In the present study, we define D65 (light under average daylight including the ultraviolet region) as a standard light, which has been defined by the CIE to determine the standard seawater color in the field. The product of the color matching function and the standard light D65 used in Equations (2)–(4) is known as the weight factor [25].

### 2.3. Discolored Seawater and the Chemical Composition Data Set

To clarify the relationship between discolored seawater and its chemical composition, the RGB values from aerial photographs and the chemical composition dataset of areas around submarine volcanoes in various parts of Japan compiled by Watanabe [18] have been used in this study. The details of the chemical analysis method for discolored seawater used in Watanabe's data are mainly based on Nogami et al. [13] and Ossaka et al. [17]. As a basic method, water samples floating on the sea surface were collected using a radio-controlled boat. Then, 1 mL of 6 N HCL was added to 100 mL of the collected sample, and the mixture was warmed in a water bath of 100 °C for 24 h to completely dissolve the precipitate in the coexisting solution. The three components of Si, Fe, and Al in this solution were quantified by the molybdenum yellow method, by the absorption photometry of 2,2′-bipyridine method, and by the atomic absorption photometry of the nitrous oxide-acetylene flame, respectively. However, this dataset obscures the statistical relationship because it contains too many of the same RGB values for different chemical composition values. Thus, to derive the relationship between the average color and the chemical composition, in this study, when the R, G, and B values were exactly the same on any given day, the Fe, Al, and Si values were changed to the average values for each day. Additionally, the dataset for Satsuma Iojima Island, where there was a period of approximately 6 months between the water sampling and aerial photography, was excluded in this study. Moreover, the component ratios of Fe, Al, and Si were calculated and expressed as Fe%, Al%, and Si%, along with ((Fe + Al)/Si) which is an index of volcanic activity proposed by Tsuchide [26]. Table 1 shows the Watanabe [18] dataset modified in this way. Then, assuming the following simple regression model, the relationship between the two was investigated using the following formula:

$$(CC) = a(OC) \tag{8}$$

where [CC] is the chemical composition (Fe, Al, Si, (Fe + Al)/Si, Fe%, Al%, Si%), [OC] is the seawater color values (R, G, B, x, and y), and a and b indicate the regression coefficient. The significance of the correlation was judged using $p < 0.01$ (the significance level of 1%).

### 2.4. Calculation of Volcanic Activity Using Satellite Data

In the present study, GCOM-C and Himawari-8 data were used to calculate the activity of Nishinoshima Island in 2020. GCOM-C is a satellite launched by Japan in December 2017 that is equipped with an SGLI sensor that observes the ultraviolet, visible, and near-infrared bands (nine bands of 380, 412, 443, 490, 530, 565, 674, 763, and 869 nm) with a 250 m resolution. In this case, since regular discolored seawater data is required, the 46 atmospheric corrected reflectance 8 day composite (8 day average) data from January 2020 to December 2020, with a 250 m spatial resolution, was downloaded from the JAXA Satellite Monitoring for Environmental Studies (JASMES) Data Access System (https://www.eorc.jaxa.jp/JASMES/SGLI_STD/daily.html?area=j (accessed on 26 March 2021))

produced by Japan Aerospace Exploration Agency, Earth Observation Research Center (JAXA EORC). The reflectance data actually obtained, 15 pixels × 15 pixels data centered on Nishinoshima Island (5 × 5 pixels) was extracted, as shown in Figure 2. The reflectance data of each pixel were converted to *XYZ* according to Equations (2)–(4). Specifically, as shown in Figure 3, the reflectance of the nine bands obtained from the SGLI data is linearly interpolated in 5 nm steps, multiplied by a 5 nm weight factor (JIS Z 8781-1). It was then converted into coordinates *X*, *Y*, and *Z*, and the converted *X*, *Y*, and *Z* values were further converted to *x* and *y* using Equations (5) and (6).

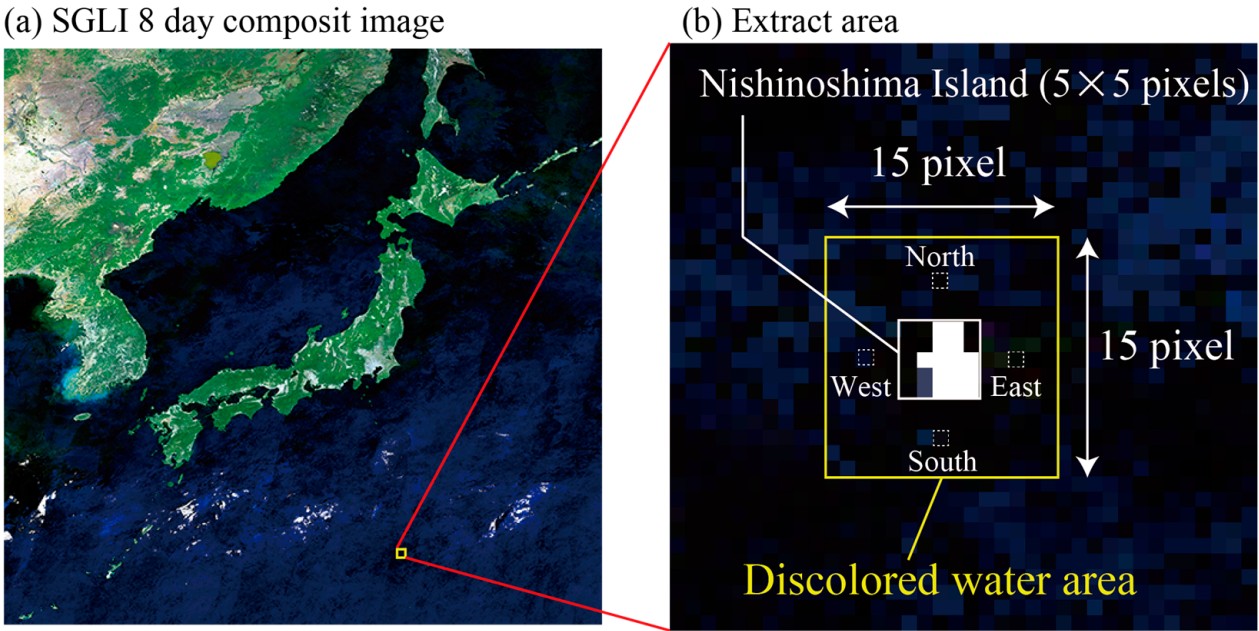

**Figure 2.** An example of the atmospheric corrected reflectance 8 day composite data of SGLI around Nishinoshima Island, Japan.

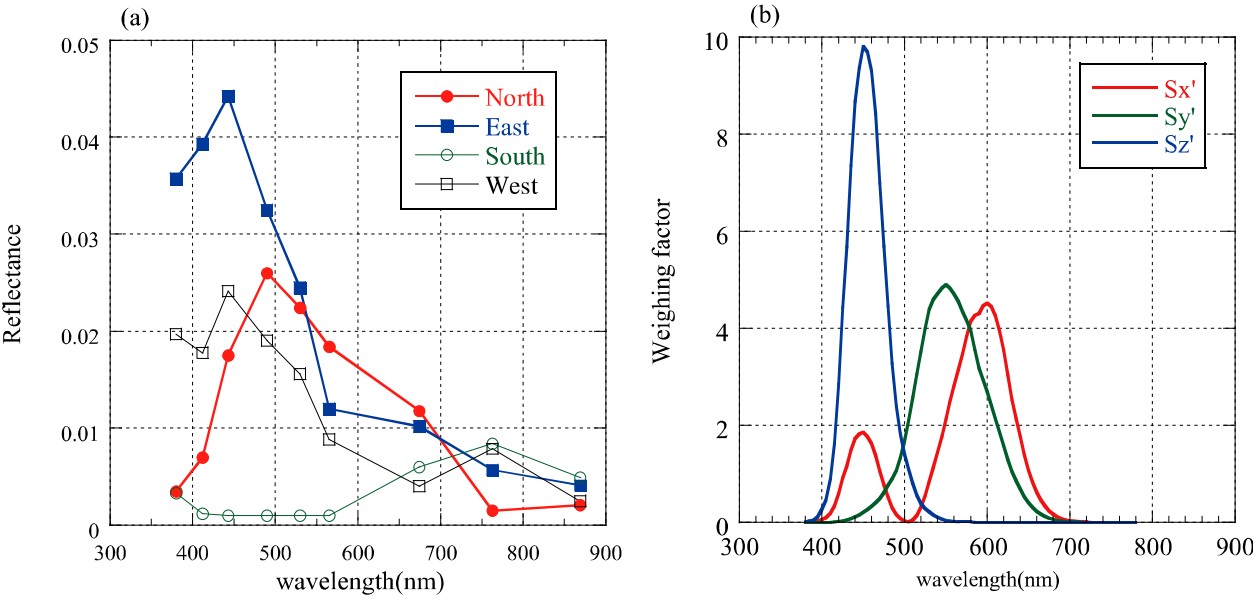

**Figure 3.** (**a**) Samples of GCOM-C atmospheric corrected spectral reflectance data, 12–20 August 2020, and (**b**) and weighting function of the CIE 1931.

**Table 1.** Modified data set of discolored seawater (RGB) from aerial photographs and measured chemical composition (Fe, Al, and Si) for discolored seawater around Japanese submarine volcanoes summarized by Watanabe [18].

| No. | Station * | Water Sampling Date | Aerial Photo Date | R | G | B | Fe (mg/kg) | Al (mg/kg) | Si (mg/kg) | (Fe+Al)/Si | Fe% | Al% | Si% |
|---|---|---|---|---|---|---|---|---|---|---|---|---|---|
| 1 | IZ | Dec.10, 1986 | Dec.10, 1986 | 112 | 146 | 147 | 0.18 | 0.8 | 1.79 | 0.55 | 6.5 | 28.9 | 64.6 |
| 2 | IZ | Dec.11, 1986 | Dec.11, 1986 | 161 | 184 | 176 | 0.02 | 0.04 | 0.24 | 0.25 | 6.7 | 13.3 | 80.0 |
| 3 | IZ | Dec.16, 1986 | Dec.17, 1986 | 161 | 168 | 174 | 1.6 | 2.75 | 6.74 | 0.64 | 14.4 | 24.8 | 60.8 |
| 4 | IZ | Dec.20, 1986 | Dec.20, 1986 | 113 | 122 | 121 | 0.57 | 1.44 | 3.09 | 0.65 | 11.2 | 28.2 | 60.6 |
| 5 | IZ | Jan. 7, 1987 | Jan. 7, 1987 | 146 | 152 | 148 | 0.36 | 1.16 | 2.25 | 0.68 | 9.5 | 30.8 | 59.7 |
| 6 | IZ | Jan. 8, 1987 | Jan. 8, 1987 | 138 | 157 | 157 | 0.24 | 1.05 | 1.64 | 0.79 | 8.2 | 35.8 | 56.0 |
| 7 | NS | Jul. 7-8, 1974 | Aug. 3, 1974 | 119 | 126 | 144 | 1.4 | 0.7 | 11.23 | 0.19 | 10.5 | 5.3 | 84.2 |
| 8 | KT | Mar.1984 | - | 221 | 220 | 200 | 0.3 | 0 | 0.22 | 1.36 | 57.7 | 0.0 | 42.3 |
| 9 | FO | Mar.21, 1977 | Jan.10, 1977 | 233 | 247 | 247 | 0.14 | 0.09 | 0.25 | 0.94 | 29.2 | 18.8 | 52.1 |
| 10 | FO | Jan.20-24, 1986 | Jan.12, 1986 | 153 | 125 | 113 | 0.54 | 0.63 | 0.56 | 2.07 | 31.2 | 36.4 | 32.4 |
| 11 | FO | Feb.1-26, 1986 | Jan.29, 1986 | 114 | 108 | 118 | 0.13 | 0.34 | 2.71 | 0.17 | 4.1 | 10.7 | 85.2 |

* IZ: Izu-Ohshima Island, NS: Nishinoshima Island, KT: Kaitoku Seamount, FO: Fukutoku-Okanoba.

The Himawari-8 is a meteorological satellite launched by Japan in October 2014 that is equipped with the Advanced Himawari Imager (AHI), which has a total of 16 bands from 0.47 to 13.3 μm. In this study, the brightness temperature in the 3.9 μm band (AHI band 7) was extracted to estimate the high-brightness temperature inside of the crater as an index of volcanic activity [27]. The projection-converted L1 gridded data with 2 km spatial resolution (https://www.eorc.jaxa.jp/ptree/index.html (accessed on 26 March 2021)) provided by JAXA EORC were downloaded. The maximum area value of 15 pixels × 15 pixels centered on Nishinoshima Island at midnight (15:00 UTC) was set, and this was used as an index of volcanic activity.

## 3. Results

### 3.1. Relationship between the Color of Discolored Seawater and Its Chemical Composition in Submarine Volcanoes

We first investigated the relationship between discolored seawater color and its chemical composition in general submarine volcanic waters. Table 2 shows the correlation coefficient between the seawater color and its chemical composition, explained using Equation (8). No significant correlation was obtained between (R, G, B) and (Fe, Al, Si, (Fe + Al)/Si). However, several statistically significant ($p < 0.01$) correlations were observed between R and Fe%, $x$ and (Fe + Al)/Si, $y$ and (Fe + Al)/Si, $y$ and Si, and $y$ and Si%, respectively. Particularly, the highest correlation coefficient ($r = 0.83$) was obtained for $x$ and (Fe + Al)/Si. Figure 4 shows the relationships between the five combinations of seawater color and chemical composition that were found to have a statistically significant correlation ($p < 0.01$). Each relationship is expressed through the following equations:

$$(Fe\%) = 0.302(R) - 28.7 \tag{9}$$

$$((Fe + Al)/Si) = 45.4(x) - 13.3 \tag{10}$$

$$((Fe + Al)/Si) = 564(y) - 17.8 \tag{11}$$

$$(Si) = -360(x) + 121 \tag{12}$$

$$(Si\%) = -1739(y) + 634 \tag{13}$$

**Table 2.** Correlation coefficient of discolored seawater color (R, G, B, $x$, and $y$) and its chemical composition (Fe, Al, Si, (Fe + Al)/Si, Fe%, Al%, and Si%).

| Chemical Composition | R | G | B | x | y |
|---|---|---|---|---|---|
| Fe | −0.17 | −0.24 | −0.14 | −0.08 | −0.45 |
| Al | −0.34 | −0.31 | −0.25 | −0.18 | −0.23 |
| Si | −0.43 | −0.41 | −0.24 | −0.4 | −0.79 * |
| (Fe+Al)/Si | 0.45 | 0.21 | 0.05 | 0.83 * | 0.75 * |
| Fe% | 0.78* | 0.59 | 0.46 | 0.61 | 0.59 |
| Al% | −0.31 | −0.29 | −0.34 | 0.11 | 0.27 |
| Si% | −0.53 | −0.36 | −0.2 | −0.67 | −0.77 * |

* $p < 0.01$.

### 3.2. Volcanic Activity Characteristics of Nishinoshima Estimated from AHI Data

Various data are required to link the discolored seawater color data from SGLI with numerical volcanic activity. The Himawari-8 AHI temperature data for Nishinoshima Island (mainly the temperature of the crater) is used as time series data for volcanic activity [27]. In this study, the time series characteristics of the temperature for the whole year (2020) were investigated by applying the method (The larger the difference between the area's average temperature (Tave) and the area maximum temperature (Tmax) in the 3.7 μm, the more intense the volcanic activity). Figure 5 shows the Tave and Tmax obtained from the AHI data. Taking Tave as the background temperature, it can be seen that the temperature was approximately 300 K (pink region) throughout the year. Additionally,

the Tmax from January to June 2020 fluctuates between 300 and 350 K (a temperature difference of approximately 50 K), and subsequently, from the beginning of June to the end of July, the Tmax increased from 350 to 400 K (an increase of 50 K). The Tmax peak occurred around July 1, and by July 29, it had dropped back to the Tave level. Afterward, the Tmax and Tave values remained almost the same, and no further increase in Tmax was observed until the end of December.

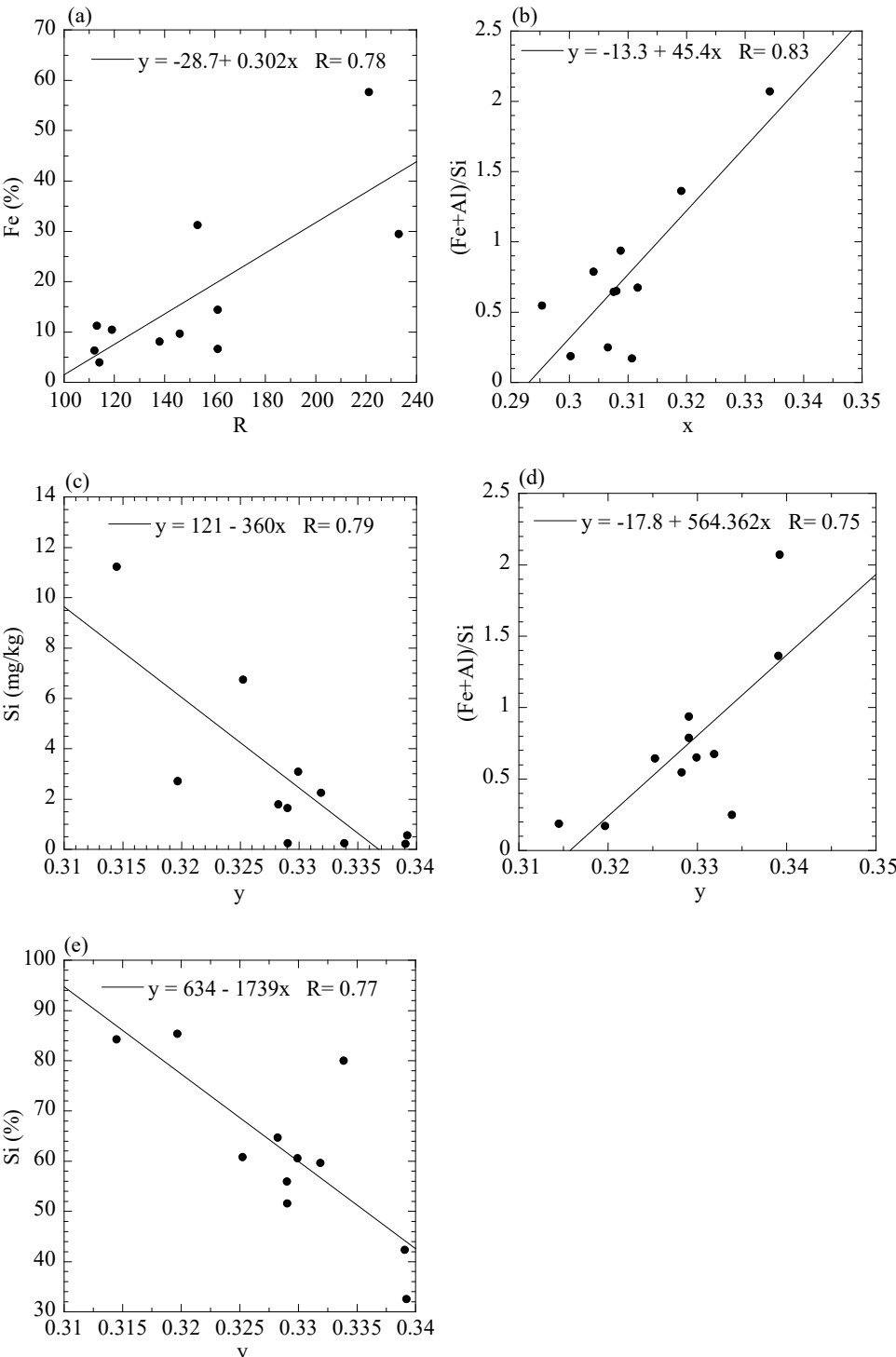

**Figure 4.** Examples of the relationship between the parameters of discolored seawater color parameters (R, x, and y) and its chemical composition (Fe, (Fe + Al)/Si, and Si). (**a**) R vs. Fe%, (**b**) x vs. (Fe+al)/Si, (**c**) y vs. Si%, (**d**) y vs. (Fe+Al)/Si, (**e**) y vs. Si%.

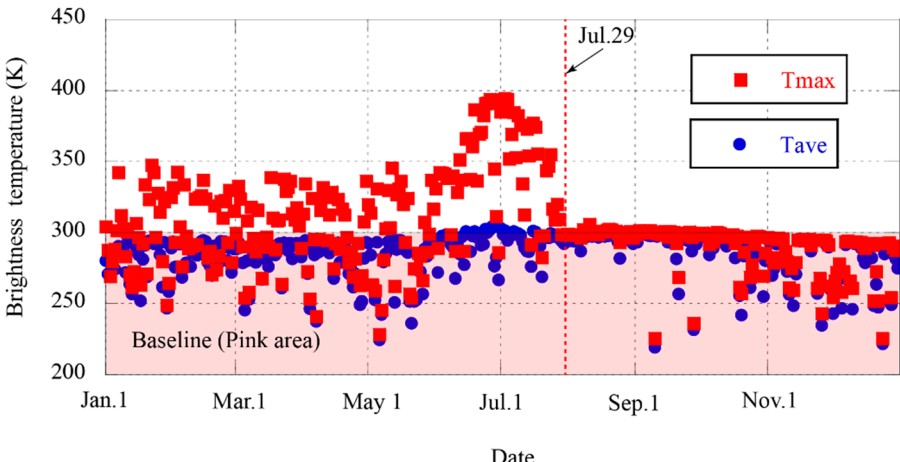

**Figure 5.** Time series of the maximum temperature (Tmax) and the average temperature (Tave) obtained from the 3.9 μm band of the Himawari-8 data around Nishinoshima Island in 2020.

Table 3 shows the activity status of Nishinoshima Island during 2020, which was obtained by the Japan Coast Guard [23], mainly through visual observations undertaken using aircraft. We have summarized the color, location, and scale of the discolored seawater described by the Japan Coast Guard in a compact manner. These data show that an eruption had already occurred by the beginning of 2020, which continued until the end of July, with the eruption subsiding for a short period during the first half of April and eruptive activity since August 19 has subsided. Hence, the activity of the volcano is in agreement with the fluctuation observed in the AHI Tmax. In other words, eruptive activity occurs when the Tmax greatly exceeds the Tave (with a temperature difference between several tens to 100 K), and eruptive activity has subsided when the Tmax and Tave values are approximately the same and below 300 K.

**Table 3.** Volcanic activity record of Nishinoshima Island obtained through visual observations by the Japan Coast Guard. (The discoloration information has been modified into a short description.).

| Date | Eruption | Discolored Water | Water Color | Discoloration Position | Discoloration Width |
|---|---|---|---|---|---|
| 17 Jan. 2020 | Yes | Yes | - | - | - |
| 4 Feb. 2020 | Yes | Yes | yellowish-brown | W/NE | about 100 m |
| 17 Feb. 2020 | Yes | Yes | - | - | - |
| 9 Mar. 2020 | Yes | Yes | yellow-green | W/N/E | |
| 15 Mar. 2020 | Yes | Yes | - | - | - |
| 6 Apr. 2020 | None | Yes | - | - | - |
| 16 Apr. 2020 | None | Yes | - | - | - |
| 19 Apr. 2020 | None | Yes | brown | E | |
| 29 Apr. 2020 | Yes | Yes | - | - | - |
| 18 May 2020 | Yes | Yes | - | - | - |
| 7 Jun. 2020 | Yes | Yes | yellowish-brown | N | - |
| 15 Jun. 2020 | Yes | Yes | yellowish-brown | E | - |
| 19 Jun. 2020 | Yes | Yes | - | - | - |
| 29 Jun. 2020 | Yes | Yes | yellow-green | All | |
| 20 Jul. 2020 | Yes | Yes | - | - | - |
| 19 Aug. 2020 | None | Yes | - | - | - |
| 23 Aug. 2020 | None | Yes | yellow-green | NW/SW | - |
| | | | Light green | N/SE | 2 km< |
| 5 Sep. 2020 | None | Yes | - | NW/SW | 2 km< |
| 28 Oct. 2020 | None | Yes | brown | SE/SW | - |
| 24 Nov. 2020 | None | Yes | brown | SE/SW | - |
| 7 Dec. 2020 | None | Yes | brown | SE/SW | - |

### 3.3. Color Characteristics of Discolored Seawater Estimated from SGLI Data

To validate whether the seawater color data obtained by SGLI accurately captures the actual conditions of the discolored seawater, we investigated the color characteristics of the discolored seawater, estimated from the SGLI data. Figure 6 is a chromaticity diagram of the seawater color at fixed points (north, east, south, and west) in the sea area around Nishinoshima Island estimated from SGLI. The numbers around the colors indicate the dominant wavelength (WL), which is an index of the hue. From this, the $x$ value of the seawater color changes within a range of approximately 0.2 to 0.5, and the $y$ value changes within a range of approximately 0.1 to 0.5. Additionally, the data are particularly concentrated in blue (where the dominant WL is approximately $475 \pm 5$ nm) from the Achromatic point (white point) to the lower left. Relatively pure green–red seawater (dominant WL from approximately 535 nm to approximately 615 nm) was also captured. There are several seawater color groups that are far from the group in zone A, such as the seawater color in zones B and C, at the north, east, and south stations. The color change of the west station is relatively small and was confirmed qualitatively.

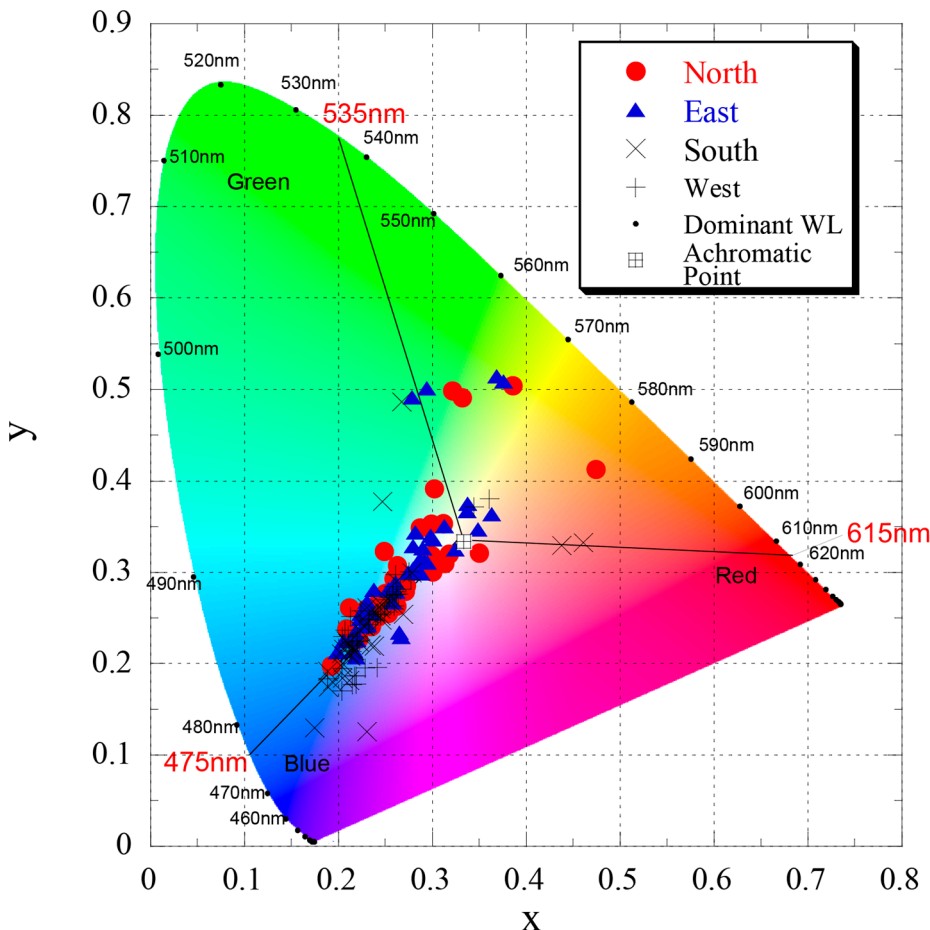

**Figure 6.** Colorimetric data of discolored seawater at the four directions (north, east, south, and west) around Nishinoshima Island in 2020.

### 3.4. Distribution Characteristics of the Chemical Components Estimated from SGLI Data

We investigated the distribution characteristics of the chemical composition of Nishinoshima estimated from the SGLI data using the estimation formula that gave the strongest significant correlation in Section 3.1. Figure 7 is an example of an estimated distribution map of (Fe + Al)/Si between mid-May and late June that has been mapped by applying Equation (10) to the SGLI data. From this, it is qualitatively well understood that the value of the northeastern part of the island rises as a whole during this period, and then

the discoloration gradually progresses to the sea area around the entire island, before it disappears.

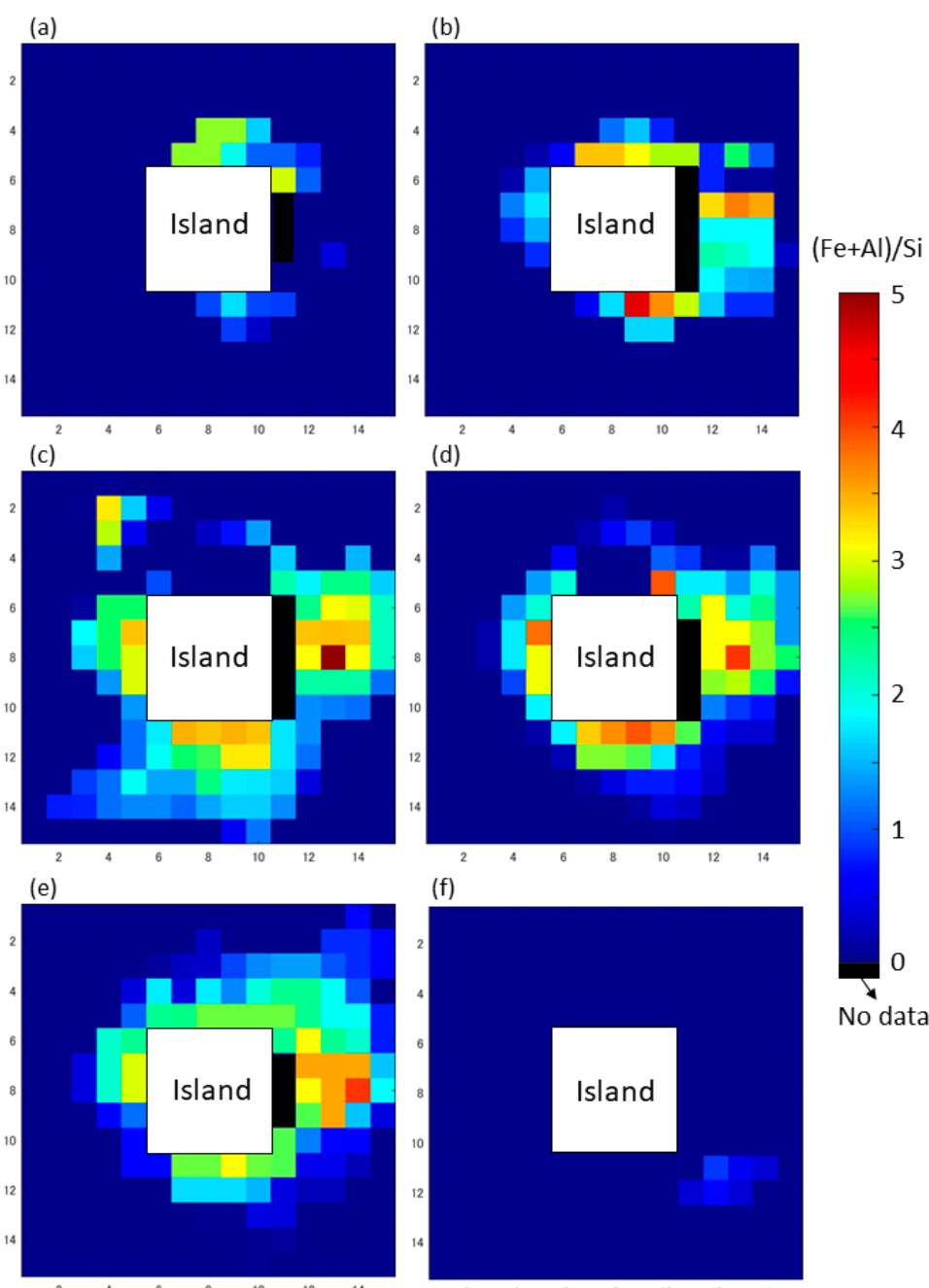

**Figure 7.** Example of the (Fe + Al)/Si distribution as a volcanic activity index from May 16 to June 25 around Nishinoshima Island. (**a**) 16–23 May 2020, (**b**) 24–31 May 2020, (**c**) 1–8 June 2020, (**d**) 9–16 June 2020, (**e**) 17–24 June 2020, (**f**) 25 June–2 July 2020.

Furthermore, we investigated the annual time series fluctuations in 2020 to quantitatively understand the changes in (Fe + Al)/Si in the discoloration range. In Figure 8, the area average (Fe + Al)/Si value of the entire sea area in the analysis range was calculated each time and is displayed in chronological order together with the Tmax (as explained in Section 3.2). Consequently, (Fe + Al)/Si fluctuated significantly in May and June, before the Tmax peak at the start of July, and then decreased in early July. Afterward, the discoloration

range decreased sharply, and it was confirmed that the value remained almost constant from the beginning of August.

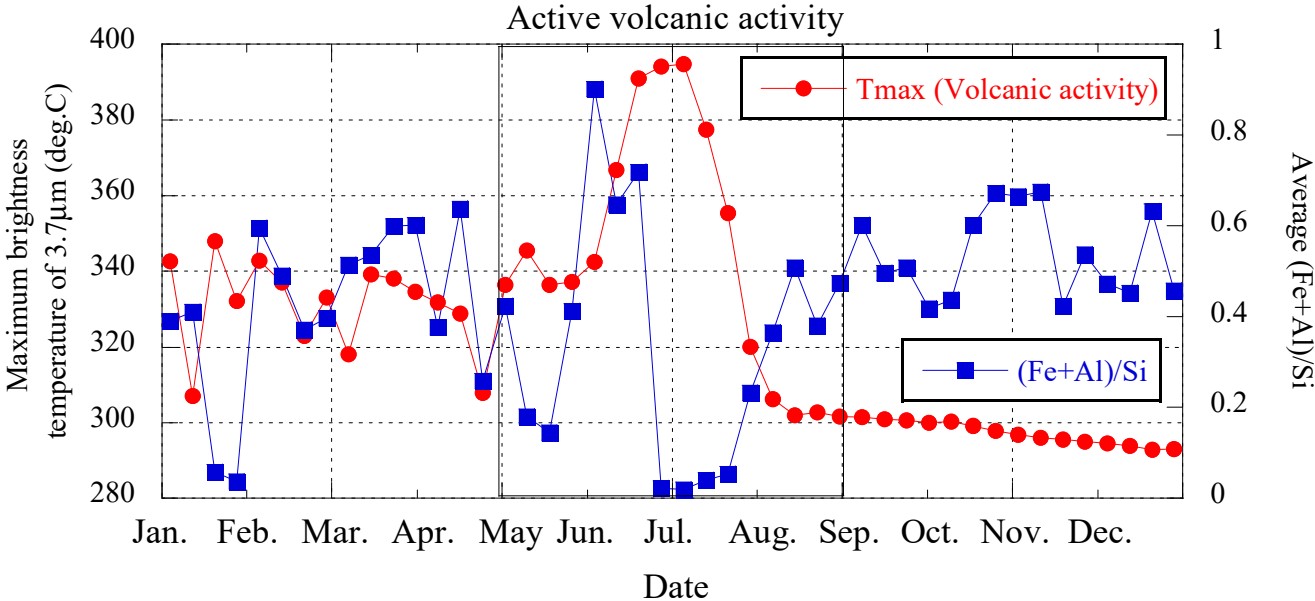

**Figure 8.** Comparison of the time series between Tmax (as a volcanic activity from Himawari-8) and an area average (Fe + Al)/Si of the discolored seawater color around Nishinoshima Island (from GCOM-C) during 2020. (Both datasets involved an 8 day cycle).

## 4. Discussion

### 4.1. Validity of the Relationship between the Seawater Color and its Chemical Composition

According to Nogami et al. [13] and Ossaka et al. [14], the discolored seawater around submarine volcanoes is known to be caused by the reaction of the $SiO_2$–$Al_2O_3$–$Fe_2O_3$–$H_2O$ system. To estimate the volcanic activity from the seawater color using this relationship, Watanabe [18] investigated the relationship between the RGB values of the aerial photographs and the chemical composition (Fe, Al, and Si) but found no significant correlation between the two. We, therefore, excluded both duplicate data and data with a large observation time gap from the Watanabe [18] dataset and reanalyzed it using the *XYZ* colorimetric system and a new (Fe + Al)/Si. Consequently, and as shown in Table 2 and Figure 4, we found a strong and significant correlation between $x$ and (Fe + Al)/Si and between $y$ and Si. Since $x$ means "red," excluding the brightness component, it has a slightly different meaning from R, for which Watanabe [18] could not find a correlation. In other words, in the correlation analysis conducted by Watanabe [14], it is considered necessary to eliminate the influence of the brightness condition due to the shooting conditions of the aerial photograph. Conversely, among the RGB values that Watanabe [18] attempted to use, the R value was found to have a strong and significant correlation with Fe% (the ratio of the three components of Fe) in this study. It is suggested that estimating the Fe% from R may be useful as a simple method for estimating the volcanic activity.

### 4.2. Validity of Seawater Color Estimation Derived from SGLI

Generally, the colors of seawater worldwide are represented using the Forel–Ure (FU) water color scale. The $x$ and $y$ of FU1 to FU21 are in the range of 0.19 to 0.47 and 0.16 to 0.49, respectively [28]. This time, the seawater color around Nishinoshima Island is within the range of FU water color in almost all of this area, and this is considered to be appropriate. However, since the SGLI data used were the composite data for 8 days, it could not be compared and validated with the visual *XYZ* colorimetric for each day. Additionally, the colors visually observed from an aircraft and the colors measured by

satellites do not always match because of the effects of weather and water surface reflection. Furthermore, although the reflectance data of SGLI has only nine bands, 5 nm step data are created by linear interpolation, and there may be some errors introduced through the interpolation method. Thus, it should be remembered that this result is based on the assumption that "the water color calculated by SGLI is correct or captures the relative change of the seawater color".

Conversely, data that changes into three systems of blue, green, and red were obtained by calculating the SGLI data. Such changes in color (or main WL) appear to be related to the chemical composition of seawater. Ohsawa et al. [19] explained, from the chemical analysis of hot spring water, that the water changes color because of the influences of Si in the blue region near the main WL of 480 nm and Al in the region near 490 nm. Onda et al. [20], after chemical analysis of crater lakes, explained that the green coloration near the main WL of 535 nm is due to the influence of $Fe^{2+}$. After consideration of these previous studies, these results are also the reason behind the change in the chemical composition around Nishinoshima Island.

### 4.3. Validity of Discolored Seawater Distribution Characteristics Based on SGLI Data

According to Ossaka et al. [17], "The chemical composition and color tone of discolored seawater change sharply in response to the fate of volcanic activity, and it is an effective index for understanding the activity status of submarine volcanoes." However, the timing of the discolored seawater and its relationship with eruptive activity have not yet been elucidated. In this study, the distribution of (Fe + Al)/Si in Nishinoshima Island estimated from SGLI data spread to the northeastern part of the island from mid-May to late-June, as shown in Figure 7. This is thought to be due to the lava deposition that spread to the northeastern coast of the island during this period, which was reported by the Geospatial Information Authority of Japan [29]. The distribution fluctuated significantly approximately one month before the volcanic activity began. By contrast, satellite analysis of the submarine volcano Fukutoku-Oka-no-Ba conducted by Urai [16] reported that there was no change in the seawater color before the eruption and that there was a large change in the seawater color immediately after. In fact, in 2020, no eruptions were observed at Nishinoshima Island after August, but discolored seawater continued to appear, and it was believed that a longer-span analysis (at least 5 years) was necessary.

### 5. Conclusions

The present study aimed to derive the relational model between the seawater color and the chemical composition in the discolored seawater area using the XYZ colorimetric system and to examine the predictability of volcanic activity in Nishinoshima Island by GCOM-C SGLI. The following three results have been obtained:

1. A significant correlation was found between the seawater color ($x$) calculated using the *XYZ* colorimetric system and the chemical composition such as x vs (Fe + Al)/Si, y vs Si, y vs (Fe + Al)/Si, y vs Si% based on the modified dataset from a previous study [18].
2. The fluctuation of the Himawari-8 maximum water temperature (Tmax) around Nishinoshima Island at midnight corresponded well with the volcanic activity.
3. The discolored seawater around Nishinoshima Island, derived from SGLI, is mostly blue at the dominant WL of approximately 475 nm, but we also observed data showing a relatively high purity of green to red at approximately 535–615 nm.
4. The distribution of (Fe + Al)/Si in Nishinoshima Island, estimated from SGLI data, fluctuated significantly just before the volcanic activity became active (approximately one month prior).

It will be necessary in the future to validate the effectiveness of the method proposed in this study. However, presently, it is extremely difficult to conduct a water sampling survey in the field when eruptive activity is active, and therefore, it will be necessary to verify the method in cooperation with institutions within the country that has jurisdiction.

Finally, we would like to perform a similar analysis on submarine volcanoes distributed around the world to elucidate the relationship between volcanic activity and the seawater color of the surrounding sea area.

**Funding:** This work was supported by JSPS KAKENHI Grant Number 17H04625,18H03731, 19H04292, 20KK0141.

**Acknowledgments:** I would like to thank Kento Hirase of Hiroshima University for helping me with the satellite data processing that is the basis of this paper. The satellite data (Himawari-8 and GCOM-C) was used in this paper was supplied by the P-Tree and the JASMES systems of JAXA. (https://www.eorc.jaxa.jp/ptree/userguide.html (accessed on 26 March 2021), https://www.eorc.jaxa.jp/JASMES/SGLI_STD/daily.html (accessed on 26 March 2021)).

**Conflicts of Interest:** The authors declare no conflict of interest.

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
