# Peer review of "Trial of Chemical Composition Estimation Related to Submarine Volcano Activity Using Discolored Seawater Color Data Obtained from GCOM-C SGLI. A Case Study of Nishinoshima Island, Japan, in 2020"

_water, doi:10.3390/w13081100_

Round 1
Reviewer 1 Report
This study aims to correlate the color and chemical composition of discolored seawater color
around a submarine volcano and to observe whether it relates to volcanic activity at
Nishinoshima Island in Japan. The author has utilized a model applied by GCOM-C SGLI
(Global Change Observation Mission–Climate Second Generation Global Imager) to meet
research objectives. The author used XYZ colorimetric system to extract chemical
composition as summarized in previous studies to correlate it with the RGB value of the
discolored seawater in the submarine volcano. A significant correlation between seawater
color and chemical composition of the discolored seawater was observed and that the
distribution of the chemical composition of the discolored seawater fluctuated just a month
before the volcanic activity took place. This research presents a novel method to
quantitatively predict the submarine volcanic activity but this method needs more validation
by replicating it in other parts of the world. The following minor corrections are suggested:
Line 11-12: It is important to mention the full form of GCOM-C SGLI dataset used in this
study
Line 15-16: Confused sentence. This must be corrected to convey clear information
Line 19: Complete name of the dataset must be used i.e., GCOM-C GLI instead of SGLI only
Line 29: It is very important to give citation to the mentioned volcanic activities
Line 39: SAR stands for?
Line 41: SiO 2 is?
Line 50-51: It is important to mention the source of this information.
Line 112-114: Need to explain what are these 0-255, are these of an 8-bit spectral value of
aerial imageries?
Line 123-124: The sentence must be corrected
Line 159: p<0.01 must be explained
Line 171: 15x15 pixel refers to which unit of measurement?
Line 181: The brightness temperature was extracted through?
Line 235-236: The parenthesis is used inappropriately
Line 239: no eruptions having what?
Line 242: and eruptive activity was not active
Line 283-286: What is the possible reason for an increase in overall value on the northeastern
side of the island? This must be discussed in the discussion section
Line 355: Use “One” instead of “1”
Line 361: As suggested that a longer-span analysis is required to observe the appearance of
discolored seawater. Does any previous study discuss for how long this be monitored? This
can be an important addition to the discussion section.
It is very important to explain the abbreviations, especially abbreviations used to mention
chemical compositions, for the reader to get the true sense of the explanation. This needs to
be addressed in the Introduction portion of the manuscript. The manuscript needs to be
carefully checked again for common grammatical mistakes, comma errors, ambiguous
sentence structures, etc. Overall, the study is interesting and presents a novel approach to
correlate the color and chemical composition of seawater around a submarine volcano. The
article can be accepted after the minor changes mentioned above.
Author Response
The response to comments is attached here.

Reviewer 2 Report
Attempts to detect submarine volcanic activity by changes in the color of seawater has high expectations for its application due to its rapidity. This study is one of such challenging attempts and is worthy of publication in Water MDPI after minor revisions as described below.
I would like to note that I am not an expert in volcanology or remote sensing, so I am unable to judge some of the technical information in such area in this manuscript. Nevertheless, in my opinion, this manuscript is logically well written and does not contain any fatal flaws. On the other hand, I consider that the following two points are major issues that need to be improved.
1) This manuscript lacks a detailed description of how the chemical composition of the water was measured. The method by which the chemical composition of the discolored water was obtained is critical to the validation of this study. For example, the reader needs to know whether the data presented as the concentration of each element indicates that only dissolved ions were measured or whether colloidal particles were also included. In addition, how the water samples were collected is also important because the temperature change of the samples from the field to the laboratory will cause precipitation of some elements.
The reference of data source (Watanabe, 2015) also did not describe the detail of chemical analysis. I found some of such information in references of Watanabe (2015), e.g., Kosaka et al. (2000). However, because all these references are written in Japanese, it is extremely difficult for non-Japanese researchers to access detailed information. Considering that this is an international journal, the authors should provide a brief summary of how the chemical data used in this study were obtained in the manuscript.
2) In conclusion part Line 367, authors argued “a significant strong correlation was found between the seawater color (x) calculated using the XYZ colorimetric system and (Fe + Al)/Si based on the modified dataset from a previous study [14]. “However, because the number of data plot is so small, the number of R=0.83 may not be enough to conclude “a significant strong correlation”. If the author has a statistical way to justify this claim, he should write about it more carefully. Even if not, statistically speaking, it is not very conclusive, but I think it is acceptable because we are dealing with ongoing volcanic activity. Just I would recommend to tone down the conclusion.
Author Response

(The authors gave the same response as above.)

Reviewer 3 Report
Review of
Trial of chemical composition estimation related to submarine 2 volcano activity using discolored seawater color data obtained 3 from GCOM-C SGLI. A case study of Nishinoshima Island, 4 Japan, in 2020 By Yuji Sakuno
Reviewer:
- E. Bickford
Syracuse University
In this paper the author presents a detailed account of attempts to us seawater color as a proxy of chemical changes around a submarine volcano. The author presents in detail the assumptions and equations used in the colorometric analysis. I confess that these techniques are beyond my expertise, but the assumptions and relations appeared reasonable to me. I accepted this review because I thought the subject would be interesting, and it certainly was! I congratulate the author for an excellent paper. In general the paper is quite well-written.
My principal comment about this paper is that I wonder why the author chose to submit it to “Water”? Yes, of course seawater is the subject, but only as a proxy for chemical changes accompanying eruption of the volcano. When I think of papers for “Water”, I think of ground water hydrology, water pollution, rock-water interactions, etc. Would not this paper be better published in another of the MDPI journals, such as “Remote Sensing”; “Geoscience”; or “GeoHazzards”? I leave this decision to the editor and the author, but I would not look in “Water” if I were looking for a paper on this subject.
Author Response

(The authors gave the same response as above.)
